# Evolution of Korean Megachurch Christianity Intensified by the COVID-19 Pandemic in a Socio-Political Context

**Kwang Suk Yoo**

Institute for Religion and Civic Culture, Kyung Hee University, Seoul 02447, Republic of Korea; ksyooii@khu.ac.kr

**Abstract:** This paper examines how megachurch congregations in South Korea responded to governmental measures to control the COVID-19 pandemic in terms of religious governance. This empirical study of *Saeronam* Church (SC) in Daejeon shows that the unexpected crisis forced its congregants to look back on their faithfulness in terms of self-reflection, and hence, made them more committed to their congregation socially and organizationally. The theologically and politically conservative megachurch congregants tend to regard the pandemic crisis as a God-planned ordeal which must be endured not only through self-reflection and repentance, but also through protection of their congregation from secular authorities. This attitude made it easier for conservative congregants to protest against governmental quarantine measures more explicitly and collectively. While some argue that the COVID-19 pandemic basically accelerated secularity by shrinking religious influence on society, this paper finds this aspect remarkably opposite in Korean conservative churches like SC, and emphasizes how a secular challenge, like the recent quarantine measures, can intensify megachurch Christianity. In this sense, it claims that the second-generation Korean megachurches like SC cannot be explained entirely by traditional theories of urbanization, marketing strategies, and church growth.

**Keywords:** Korean megachurch; Korean Protestantism; COVID-19 pandemic; religious conservatism; *Saeronam* Church

## 1. Introduction

The nationwide popularity of megachurches in Korea is a historically and culturally unexpected phenomenon that originated when American missionaries first began to enter the Korean peninsula at the end of the 19th century, much later than they arrived in Japan and China. The number of megachurches with more than 2000 attendants has steadily increased in Korea over the last forty years, amounting to about 900 in 2020 (J. Kim 2020), even though the proportion (or percent) of the population belonging to Protestantism scarcely changed, remaining around 19% for the same period. The COVID-19 pandemic made both quantitative and qualitative disparities between megachurches and small or medium-sized churches more remarkable and permanent. In this context, scholars of Korean megachurches now need to pay more attention to their contextual and unique situation rather than simplifying the Korean megachurches as a monolithic unit of analysis. The simplification reduces the phenomenon of megachurches to a byproduct of secularization, or a religious ethos of social and economic modernization necessary for a developing country. Given that megachurches have frequently been criticized for issues such as the succession of ministry leadership by family members, leaders' financial corruption, and collective struggles for church power, they were also regarded as a main reason that many Christians and non-Christians alike stayed far away from the Korean religious market (Shin 2010).

Unlike such unilateral simplification of Korean megachurches, however, this paper tries to investigate their dynamic change and features as socio-political actors, focusing on an intensified Christianity of megachurch congregations as shown in their remarkable responses to the pandemic crisis. *Saeronam* Church (SC) in Daejeon which was selected as

the main target of this case study to infer a general pattern of megachurch congregations, given that megachurches located in the Seoul metropolitan area are relatively well known in the public. The city of Daejeon, in the middle of South Korea, has functioned as the key stronghold for the development and education of high-tech or advanced science, which surrounds SC geographically. The socio-historical background of Korean Protestantism also had an important impact on maintaining the megachurch Christianity of SC as a religiously cohesive community during the COVID-19 pandemic period.

In addition, this paper focuses on a unique feature of *Hapdong*, the biggest Presbyterian denomination of Korea, even though many other Protestant megachurches in Korea belong to various denominations such as Methodism, Baptism, or Pentecostalism as well as Presbyterianism. This denominational feature of SC is regarded as another factor that explains how their Christianity intensified during the COVID-19 crisis, as shown in their civil movements to protest against governmental quarantine measures, as well as their theological attitudes toward the COVID-19 pandemic. Its chief pastor, Rev. Jung-ho Oh, did not hesitate to lead a nationwide Protestant-based civil movement, called *Yejayeon* (Free Solidarity of Citizens for the Restoration of Worship), which eventually filed a collective suit against the Korean government for infringing on citizens' religious freedom with the governmental control measures used to prevent the spread of COVID-19. This unexpected conflict between mainstream Protestant churches and the government, controlled by the progressive ruling party, caused scholars to think that a new relationship would emerge between the two due to the current or upcoming pandemic crisis. Finally, it explains how the COVID-19 pandemic crisis has been utilized to strengthen the religious commitment of megachurch congregations, to maintain a relatively sound financial structure, and to guarantee their religious freedom more thoroughly, contrary to superficial expectations that all religions will shrink from financial losses and governmental control measures caused by the pandemic policy (Choi 2021, pp. 73–76).

## 2. Social Background: From a Myth of Growth to Public Enemy

It is generally known that megachurches, as a new social phenomenon, emerged in the 1970s and began to receive intense attention from mass media since the 1980s in the USA (Eagle 2015, pp. 590–1). Similar to this religious trend in America, it was in 1973 that *Yoido* Full Gospel Church (YFGC), with 480,000 registered memberships currently, opened its new building, capable of holding 12,000 members simultaneously, on the island of *Yoido*, where the National Assembly of Korea is located. Since then, YFGC has become a role model of Korean Protestantism, which would pursue physical growth and geographical expansion at both the local and the global level. Currently, Korea is known as the nation which shows the most remarkable growth and prosperity of megachurches in Asia.

However, this special popularity of megachurches in Korea has been entangled with their weaknesses, revealed by both mass media and scholars of various fields, especially politics, sociology, religious studies, feminism, and even theology since the 2000s, when Korean Protestants decreased greatly in the 2005 demographic census. While megachurches were regarded as symbols of the explosive growth and prosperity of Korean Protestantism in the past, now they are apt to be stigmatized as an untouchable hotbed which fosters various kinds of corruption, like the embezzlement of church money, sexual exploitation, imperial pastors, family succession of the ministry, collective conflicts for church power, and so on. They looked like "an invention of greed" (Marty 1990); a byproduct of secularity under the mask of religion derived from urbanization and consumerism (Gauthier et al. 2013, p. 5), or, at most, an out-of-style religious fashion to disappear soon. Thus, many pastors and scholars in Korea also took it for granted to put the blame for an increasing public antipathy against Protestantism on these negative effects of megachurches (J. Kim 2020; Shin 2010).

Above all, the 2016 Candlelight Revolution, which forced former president Park Geun Hye to step down, very clearly showed a social and political confrontation between anti-Protestant and pro-Protestant civil groups in Korea. Her conservative regime was brought

down by a large-scale political demonstration which two-thirds of the population participated in, finally putting both her and her political aides in jail. However, this peaceful revolution left behind an irreparable mortem religiously as well as socially and politically, because both anti-protestant and pro-protestant civil groups were heavily engaged in the politically sensitive issue of impeaching the former president Park, who was strongly supported by mainstream Protestant churches, including megachurches. Given that Korean megachurches, by denomination, have had a considerable impact on mainstream Protestant churches socially, politically, and even theologically by supporting church associations, seminaries, and overseas missionaries financially and organizationally, it seems that the success of the 2016 candlelight revolution meant a serious challenge to their social and political hegemony. As a result, the rule of a progressive regime led by Moon Jae-In began in 2017 under high tensions with conservative Protestants, then leading his ruling party to an overwhelming victory in the general election in 2019.

This drastic change in the social and political environment before the first official report of the COVID-19 infection in Korea on 20 January 2020 is closely related to the way Korean megachurches interpreted and responded to the pandemic. It meant that the long-term honeymoon period between megachurches and the ruling regime could not last any longer. In fact, the new progressive regime, underpinned firmly by the Korean Federation of Trade Unions (KFTU) and numberless civil movements groups, did not hesitate to impose heavy regulations on real estate holders and raise the minimum wage legally. However, its failure to create an integrative solidarity paid a large price socially and politically, causing conservative Protestants to launch a tenacious civil resistance, called *Taegeukgi* rally and led by the Christian Council of Korea (CCK), that most Presbyterian megachurches, including HC and SC, belonged to at that time. Rev. Kwang-hoon Chun, chief pastor of Sarang *Jeil* Presbyterian Church (SPC), played a crucial role in triggering a high level of social and political conflict with the progressive regime through the *Taegeukgi* rally, and, hence, he h underwent imprisonment and probation continually. The long-standing conflict between the progressive regime and conservative Protestants has become a potential obstacle to Korean society in coping with the COVID-19 pandemic crisis nationally.

## 3. Megachurches as a Stronghold of Faith

At the beginning of the COVID-19 pandemic in February 2020, it was *Shincheonji* Church of Jesus (SCJ) that reignited the anti-Christian sentiment of Korean people when it was reported by the Korean government and mass media that forty-two members of SCJ had traveled from the city of Wuhan, the origin of the coronavirus, and eventually began to spread the infection throughout the city of Daegu and other cities in Korea (Hancocks et al. 2020). It seemed that both the progressive regime and mass media were heavily bent on ascribing all social and political responsibility for the influx and spread of the coronavirus to SCJ, one of the most threatening sects against mainstream churches because of its aggressive missionary strategy (Yoo and Suh 2022, pp. 5–6). Since most mainstream churches, including SC, had tried to prohibit SCJ members from entering their church building in private or public, they remained on the sidelines when the progressive regime shut down the emerging sect and arrested its leaders.

However, the relationship between the government and megachurches changed drastically when many additional churches' services were reported to be a medium for spreading the infection nationally. When *Wangsung* Church in June 2020, *Ilgok Jungang* Church in July 2020, and *Woori Jeil* Church and SPC in August 2020 caused a mass infection, called the second round of the COVID-19 pandemic, the government first ordered an entire ban on face-to-face worship service in the Seoul metropolitan Area on 19 August 2020, and then across the nation (Woo 2020) Then, with the occurrence of mass infection within *Jesusvision* Holiness Church in December 2020, Back to Jesus Mission Center, International Mission Center, and *Jinju* Prayer House in January 2021, the government announced the prohibition and reduced permission for face-to-face worship (Shin 2021). During the whole period of the COVID-19 pandemic, Protestant churches, prayer houses, and mission centers across

the country had to be monitored and controlled intensively by local and central authorities in the name of preventing collective infection and spread. Frequent reports of church infection caused by violations of quarantine measures had a negative impact on how the government should handle the more than 900 megachurches across major cities to calm the pandemic, whereas Buddhist temples and Catholic churches complied patiently with ever-changing governmental measures since the time SCJ members were charged with bringing and spreading the COVID-19 virus from Wuhan in China to Daegu in Korea in March 2020. Korean Catholicism has always worked to create a cooperative relationship with progressive regimes led by Catholic presidents like Kim Dae-Jung and Moon Jae-In. For Korean Buddhists, it is not a religious duty to visit or gather in temple regularly, but rather a much more important duty not to make a conflict with secular authorities.

The compulsory ban on face-to-face worship service and activities amplified an implicit antipathy between politically conservative Protestants and the progressive regime, eventually causing the former to resist the policies of the latter, such as requiring wearing a mask and preparing an entry list. Individually, *Segero* Church in Busan which belongs to one of the most conservative Presbyterian denominations, continued to have a face-to-face service in the churchyard, keeping social distance, when its church building was shut down compulsorily by the government. Their more organized and collective resistance was revealed in *Yejayeon* (Free Citizens' Solidarity for Restoration of Worship), which was led by Rev. Jin-hong Kim, one of the most extreme rightist Presbyterian pastors and a strong partner of Rev. Kwang-hoon Chun, in making a collective demonstration against the former president Moon Jae-In. The newly organized *Yejayeon* was supported by about 1500 churches across the country, including major megachurches in Seoul such as Yoido Full Gospel Church, Sarang Church, and Myung-Sung Church, and recently won an administrative case against the compulsory ban on face-to-face worship service on 10 June 2022 through the decision of the Seoul Administration Court stating that the government does not have the right to shut down a church in the name of preventing collective infection if the church followed a few necessary measures such as mask wearing and preparing an entry list (Song 2022).

As a result, President Moon Jae-In's progressive regime had to find a political way to compromise with religious leaders, especially Protestant church leaders, because the approval rating for President Moon was plummeting steeply in the polls. In September 2020, the government launched an association for quarantine policy between government and religious representatives, and then invited leaders of seven main religious bodies to the Blue House, the presidential office (H. Kim 2020). However, this change in the ruling politicians' attitudes toward religious powers seemed too nominal and arbitrary a gesture to bridge deep policy conflicts between the two powers. This association, nonetheless, was regarded as the only channel for forming cooperative governance between progressive political and conservative religious powers, in that Protestant church leaders pleaded with President Moon and his bureaucrats for better awareness of what worship service means religiously to them. On the other hand, this meant a failure of governance to manage and control the pandemic crisis effectively. The failure was unavoidable because Moon's regime, created by the Candlelight Revolution in 2016, the largest scale of civil resistance in Korean history, succeeded in passing a few reformative laws that threatened the economic foundation of the middle and upper classes, especially multi-homeowners and market-dominant companies, who have formed a new conservative coalition by means of megachurch congregations in the Gangnam and Gangdong districts of Seoul and Bundang New Town, adjacent to the districts since the 2000s (J. Kim 2020, pp. 113, 199).

These newly grown Presbyterian megachurches like *Onnuri* Church (OC), *Sarang* church, *Myung-Sung* Church, and *Oryun* Community Church (OCC) have built a Korean version of a Bible belt across the richest districts in Seoul, which function as a solid political foundation for a conservative party regardless of religious affiliation. Their socio-politically conservative attitude has clearly been revealed in public, while at the same time keeping distance from overt political activism. However, once the COVID-19 pandemic happened

it became much easier for conservative megachurch congregants to resist organizationally and collectively against partly radical, and sometimes impromptu, policies of the progressive regime including the quarantine policy. As the pandemic lasted even longer than expected at the beginning, and the government could find no alternative but to raise or lower the level of quarantine every two weeks, both conservative and progressive citizens were rapidly losing faith in the validity of its policy. Whereas small churches and small business owners were driven into financial bankruptcy without a sustainable subsidy or systematic support program, these second-generation megachurches still function as market rulers free of competition, whose authority was intensified by the pandemic crisis both religiously and socio-politically.

Rev. Jung-Ho Oh, chief pastor of *Saeronam* Church in Daejon, and one of its elders described the situation in a similar way:

> "The governmental measures such as numerical limitation in attending worship services and prohibition of all meetings with more than five persons do not inflict serious damage on financial condition of our church by God's grace. Rather our church tends to spend more of our budget on aiding needy neighbors and missionaries abroad than before. My real concern as a pastor is in finding how church members can keep Christian values and faith strong even by attending worship service online". (Interview with Rev. Jung-Ho Oh, 22 August 2021)

> "There is no problem in financial situation of our church because we do not need to spend a large amount of expenditures on various kinds of church activities and programs, even though a certain amount of offering and devotion reduced clearly. We Christians came to realize the importance of face-to-face worship and various meetings among congregants through an unexpected ordeal of governmental quarantine measures such as limitations in church activities. I believe it occurred within God's plan". (Interviewed on 29 April 2022)

## 4. New-Born Megachurch and Theoretical Implications: A Case of *Saeronam* Church

*Saeronam* Church (SC) founded in 1986 in Daejon is a good example of understanding how Korean megachurch Christianity intensified during the pandemic crisis. Like its new partners in the nearby Gangnam district in Seoul, SC has grown so rapidly since the 2000s that it moved to its new church building adjacent to the new middle-upper class residential area, called Gangnam, in the city of Daejon. Its building, with a imagery of giving birth, matches with many modern-style commercial buildings around it, embedding a meaning of 'being new-born' to the scientific techno-valley area which is known for a well-maintained and pleasant residential district, crowded with high-income and well-educated researchers, educators, legal professionals, public officials, and IT developers (Oh 2016, vol. 3, p. 122). Rev. Jung-Ho Oh has served as chief pastor since 1994, and then led its congregation to rapid growth in membership. It is noteworthy to listen to his story and philosophy behind such rapid growth:

> "When I became its chief pastor in 1994, this church had not a few problems such as internal division among about four hundred congregants, discords among elders, and unsystematic faith-training program. One year later I introduced a discipleship program taught by Rev. John H. Oak, who is my mentor. Now all elders of this church are graduates of the discipleship program and are very modest enough to respect each other in depth, not to mention a faithful confidence in and support for me ... The most serious challenges triggered by the COVID-19 pandemic are congregants' secularization and anti-Christian attitudes of the government. We Christians have not only individual duties, but also social ones for local and national community. Thus, I preach to church members to practice Christian values in their daily life. This is a spiritual war unavoidable in this world full of lots of temptation". (Interviewed with Rev. Oh on 18 September 2022)

In fact, Rev. Oh is one of the most active pastors of Korean megachurches which struggle against enacting anti-discrimination laws to prevent discrimination in employment, education, and vocational training of educational institutions due to gender, disability, medical history, age, sexual orientation, country of origin, race, skin color, language, etc. without reasonable cause. Since the former president Roh Moo Hyunn's progressive regime first proposed the law in 2007, seven bills have been introduced until the 20th National Assembly but failed to pass. The General Assembly of Presbyterian Churches in Korea (GAPCK), called *Hapdong* ( 合同), which SC belonged to, is one of the largest Protestant denominations and simultaneously raises the most conservative voices in political and social issues, as well as a theological objection to ecumenical movements. Rev. Oh was elected as the vice-president of GAPCK in September 2022, and hence became more engaged in a 'spiritual war' against the 'secularizing' of Korean society and culture. Through the victory of the war, Christians, he thinks, can and must be 'new-born', as the word '*saeronam*' means literally. This fundamentalist approach of SC was demonstrated in a lawsuit in which Rev. Oh was charged with defamation of a Protestant sect, called *koowonpa*, led by Ock Soo Park in 2006. At that time, he led a church-based civil group to respond to the missionary work of *koowonpa* in the city of Daejeon, and let the civil group distribute about 300,000 flyers to announce the heresy of the sect in public. After a few years of judicial struggle, he was finally acquitted of the defamation charge and still remains proud of his response to the sect. It seems that he strengthened his church leadership by means of his strict attitudes toward new Protestant sects. Theoretically speaking, how Korean megachurch congregants, including the SC congregation, responded to the pandemic crisis and governmental policy towards is too complicated of a matter to be reduced to a simple and one-sided level, because their socio-political needs as well as theological tendencies are highly diverse and depend on their individual or collective situations. While their responses to the former are related mainly to the matter of how they legitimize the reaction to the pandemic crisis theologically, the latter is more closely related to what extent they rely on secular authorities that have restricted religious freedom in the cause of national security or scientific quarantine. In this sense, both aspects should be considered in the Korean context in detail.

First, the COVID-19 pandemic crisis provided them with empirical evidence and the experience of upgrading their faith in eschatology, which can open up new opportunities to think over their individual or collective commitments on one hand, and their social responsibility on the other hand (Lee 2022, pp. 61–62; MDI 2022c, p. 13). In this context, recent surveys of Korean Protestants show that the percentage who have come to believe that we human beings do not live in a sustainable civilization, or that conserving the earth is a divine duty imposed on all Christians, rose remarkably during the pandemic period (Kim 2021, pp. 805–6; MDI 2022a, p. 10). According to the surveys of *Mokhoi* Data Institute (MDI), Korean Protestants, including megachurch congregants, relied on their individual meditation and prayer most heavily (58%), along with the preaching of their chief pastor (48%), and fellowship with cell-group leaders and members (18%) to keep their faith during the pandemic period (MDI 2022b, p. 4). This implies that unlike the general expectation of weakening Christian commitments by stopping face-to-face religious services, the pandemic crisis in Korea had a much greater impact on the method of supplying and consuming religious services rather than on the extent of religious commitment. As of August 2022, Korean churches are recovering attendance, devotion, cell-group meetings, and other services very rapidly back to pre-COVID-19 levels, and commonly try to find a new model of church and preaching in accordance with the post-pandemic era.

Second, it seems that a new niche market has grown remarkably in Korea with an explosive demand for online religious services accelerated by the pandemic despite this rapid recovery. Figure 1 shows how frequent the Korean words meaning worship service have been used for searching on YouTube, where the terms "*yebae*" referring to Protestantism, "*buphoi*" to Buddhism, and "*Misa*" to Catholicism. Since February 2020, when the COVID-19 pandemic began in Korea, the number of searches for *yebae* has overwhelmed that of searches for *buphoi* and *misa*. This suggests that Protestants in Korea were much

more eager to join worship services online than Buddhists and Catholics during the pandemic crisis. This trend is still maintained in the post-pandemic era, showing that the online religious market has become more popular and interactive than before for Protestants. Now, a supply and demand of religious services online came to be an effective and even unavoidable option, especially for both megachurches and their members, in that Korean megachurches have utilized digital media and spent considerable budgets to manage their congregation and branch churches across the nation in a more organized manner.

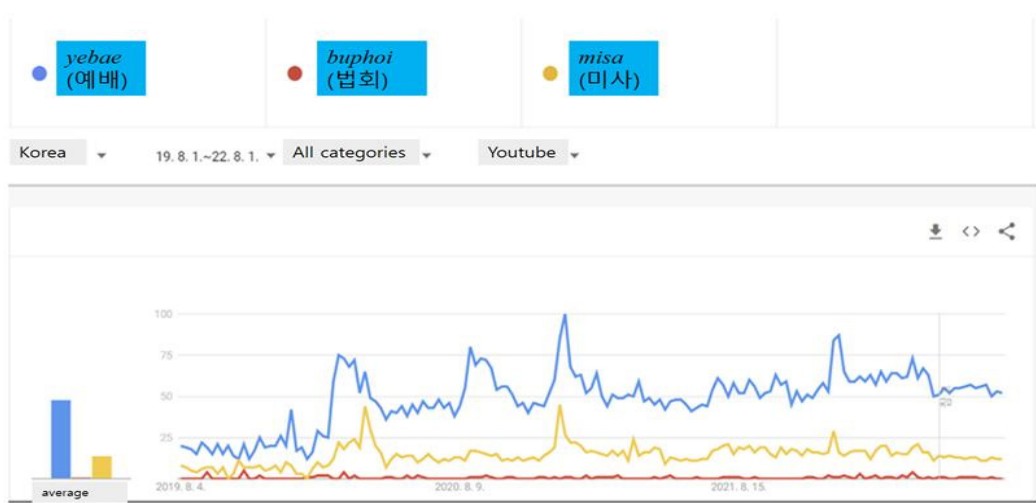

**Figure 1.** Change in Google Trends of Worship Services in Korea.

Especially during the pandemic crisis, the online religious market of Korea has grown rapisdlyt and eventually produced a few overwhelmingly successful winners such as *Bundang Woori* Church with about 260,000 YouTube subscribers, Daniel Prayer Community operated by *Oryun* Community Church with about 290,000, Good Shepherd Methodist Church with about 220,000, and so on. While these 'new-born' megachurches become even more outstanding and influential in the online religious market than in the traditional religious market, the traditionally well-known megachurches such as YFGC and *Kumnan* Church, known as the largest Methodist church in the world, have difficulty recruiting in the online religious market. In this context, the COVID-19 pandemic crisis looks like an unexpected turning point in creating new winners and losers in the online religious market, which is expected to trigger a new structural change of the Korean religious market, probably from the oligopoly to "covenant pluralism" (Stewart et al. 2020, p. 1). This is why all religious suppliers are basically exposed to a high level of free competition in online religious market which works independent of the traditional factors of religious choice such as denominational affiliation, scale and requirements of registered membership, even the presence or absence of faith or religious affiliation.

Third, from the perspective of religious policy, Korean society needs a new model of relationship or "religious governance" between Korean megachurches and the government. A deteriorated relationship between the two during the pandemic period imposed another social condition on both megachurch congregants and politicians who are willing to be engaged in highly sensitive socio-religious issues like animal rights and anti-discrimination law, whereas conservative megachurches consistently supported the anti-communist policy of dictatorship regimes and mutually promoted their common socio-political interest in the past. This also requires scholars to conduct more cross-cultural surveys of Korean megachurches. For example, whereas the first-generation Korean megachurches like YFGC were influenced by black American Pentecostalism, its political attitudes were very different from its American partners. For example, it has shown a socio-political position much closer to white megachurches in the south of America rather

than black American megachurches, which did not participate so actively in political issues (Bauman 2022, pp. 119, 141).

Fourth, the still growing second-generation Korean megachurches such as SC are not simply a by-product or inevitable phenomenon of urbanization. It is not difficult to find certain prototypes of megachurches in the history of Korean Protestantism long before the beginning of Korean urbanization and industrialization in the 1960s. Although it is well known that *Jangdaehyeon* Church, established in 1893 in the city of Pyongyang, had about 1500 seats, the actual number of attendees was always more than that (Yang 2008, p. 125). If we are allowed to define 'megachurch' in a more historically and cross-culturally flexible context, it cannot be explained comprehensively by a traditional perspective on megachurches which assumes that urbanization results in the megachurch phenomenon in Korea as well as America (Kim 2014, p. 264; Thumma and Bird 2015). David E. Eagle argues that "they represent an enduring model of ecclesial organization in Protestantism, stretching back to the early 17th century" (Eagle 2015, p. 602). In his book, *Cities of God*, Rodney Stark also proves through historical hypotheses that "larger cities had Christian congregations sooner than smaller cities" (Stark 2006, p. 81). This affinity between Christianity and cities was seen in the cities of the Roman empire, and will be seen in the future. In addition, the Redemption Camp built by the Redeemed Christian Church of God outside Lagos, Nigeria is another interesting example to consider the relationship between megachurches and megacities from a cross-cultural viewpoint. In fact, it functions as an entire city with two big auditoriums, twenty-five megawatt power station, a university, a polytechnic college, a bible college, water plant, banks, health center, a clinic, a park, guest houses, resort, secretariat, post office, international guest house, eateries, markets, offices, shops, estates, and the like (Jonah 2022). This shows that a megachurch has constructed a city, in contrast with the basic assumption of the traditional perspective. Of course, urbanization can be understood as a geographical or cultural context in which megachurches grow or decline, but not as an indisputable determinant to explain the creation, maintenance, and extinction of the megachurch phenomenon comprehensively.

Finally, a case study of *Saeronam* Church requires scholars to pay more attention to a rationalistic aspect of megachurch Christianity which has been ignored very frequently. The unexpected growth of second-generation megachurches like SC in Korea neutralizes the explanatory power of the traditional church-growth theory that focuses heavily on outdated marketing skills of megachurches like McDonaldization, consumerism culture, bureaucratic systems, or personification as Hong (2003) explained. Unlike the bureaucratic cell groups of YFGC examined by Hong (2003, p. 245), however, layperson leaders of about five hundred cell groups of SC are not only well-educated intellectuals reflecting a regional specificity surrounding SC, but also orthodox Presbyterians who value a rational and restrained self-reflection over a Pentecostal enthusiasm. Thus, it does not make sense that SC congregants are regarded simply as consumers who lack discernment in making a religious choice due to the pastor's charismatic leadership or church marketing. An elder of SC tells his experience of church life in this context:

> "I retired recently after working for thirty years in the Korea Research Institute of Standards and Science (KRISS). When I was a PhD student in USA, I, as a scientist, used to go to church because of my wife's favor without a serious belief in God. Even when I returned to Korea with getting a stable job from KRISS in Daejeon, I never realized the meaning of church or faith in my life even though my family joined SC since then. However, my life began to change little by little through joining the discipleship program of our church led by the pastor Oh. The discipleship program provided an important chance for me to understand the Christian meaning of my life more rationally". (Interviewed on 18 September 2022)

This kind of rationality within the demand side of second-generation megachurch congregations, belonging mostly to Presbyterianism, resembles the general process of making a rational choice, and hence rationalizes its own megachurch Christianity. If it is admitted

that a megachurch is not simply a product of random or secular religiosity on the demand side, further research on the rational aspects of its demand side, not the supply side, are crucially required to reach a balanced understanding of the variety of megachurches which continue to grow, even in the contemporary world.

## 5. Conclusions

Unlike first-generation Korean megachurches, which had formed a political alliance with the ruling regimes to promote anti-communism and modernization, their second-generation partners have grown recently since the 1990s and have been more engaged in strengthening the socio-political foundations of its congregants, called "conservative rationalism". The recent full-scale conflict between the progressive ruling regime and conservative Protestants, including many megachurch congregations, clearly showed how the basic principle of the separation of religion and state can be applied in different ways and sometimes nullified in terms of religious governance. Especially during the pandemic period, it was revealed not only that the eschatological crisis can lead Korean Protestants, like SC congregants, to a higher level of religious commitment through religious self-reflection and confession, but also that their Christianity can start an all-out engagement in a secular civil society in the name of religious freedom. As mentioned above, an arbitrary ruling of the former progressive regime, which thoroughly ignored the democratic process of religious governance, can be regarded as one of the main reasons why it lost the 2022 presidential election in Korea. The unexpected victory of the conservative regime implies that the socio-political influence of Protestants, especially megachurch congregants like those who attend SC, still remain active in Korea and furthermore, their Christianity can be politicized to win a 'spiritual war' against secular challenges at any time.

Conclusively, the middle class-oriented Korean megachurches like SC, which are outstanding in the region south of the Han River, function as a much more significant case to theoretically understand the rapid growth of 'rational conservatism' and megachurches in Korea. In addition, it begs a few simple and basic questions: What do megachurches mean to the Korean middle classes? How and why are they different from their domestic or international partners in terms of religious governance? What is their future in a secularizing Korean or global civil society? For a better understanding of Korea and the global megachurch phenomenon through answering these questions left behind, more continual and intensive research on megachurches is required.

**Funding:** This research was funded by the National Research Foundation of Korea [NRF-2021S1A5 C2A02088321] and John Templeton Foundation ('Megachurch in the Global South' Project).

**Institutional Review Board Statement:** The study was conducted in accordance with the Declaration of Helsinki, and approved by the Institutional Review Board (or Ethics Committee) of Canisius Colledge (protocol code 2021-22#23; 14 November 2021).

**Informed Consent Statement:** Informed consent was obtained from all subjects involved in the study.

**Conflicts of Interest:** The author declares no conflict of interest.

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
