# Peer review of "Evolution of Korean Megachurch Christianity Intensified by the COVID-19 Pandemic in a Socio-Political Context"

_religions, doi:10.3390/rel13111109_

Round 1

Reviewer 1 Report

The article makes a needed contribution to studies of megachurches during the pandemic period. On the other hand, the structure of the paper would have to change with an increased focus on the subject of the actual case study (i.e., Saeronam Church). In other words, the author would do better to lengthen the section on Saeronam Church and shorten the previous paper sections for concision and clarity.

At times, the writing becomes heavily dense and almost impenetrable; a reader, for instance, would struggle to go through the sentence "In this context the study of Korean megachurches now requires scholars to adopt a different perspective from the way of simplifying the entire Korean Protestantism as a unit of analysis by which the phenomenon of megachurch is reduced to a byproduct of secularization or a religious ethos of social and economic modernization necessary for a developing country" beginning at about line 31. Expressions such as "integrative supra-partisan solidarity" (line 119) seem incredibly confusing and jargonesque, and the writer ought to try to consider breaking down this overly abstruse language to accommodate the sensibilities of an audience that might have an understanding closer to that of an intelligent scholar (and not some theoretical specialist).

On lines 155 to 160, the author may wish to consider a slightly fuller explanation on how Catholic Christian and Buddhist houses of worship continued to work more freely according to the tacit permission of local authorities, given the longstanding positions of Catholics and Buddhists in accommodating the state and avoiding conflict whenever possible. The author can then reemphasize this explanation around lines 306 to 315.

As a scholarly courtesy to the reader, the author should consider producing an English translation of the Korean annotation of the graph at around line 300.

Author Response

I really appreciate your suggestions for correcting English words and sentences. I would be grateful if you give more suggestions for my revised manuscript, even though it reflects almost all your suggestions literally. 

Reviewer 2 Report

I found some awkward English which should be corrected before publication. See attached file

Author Response

(The authors gave the same response as above.)

Reviewer 3 Report

It is not entirely clear what the thesis of this article is. In the introduction, the author writes that the paper will explore how Korean megachurches function as socio-political actors. This is touched on again in the conclusion when it is noted that the progressive regime lost the 2022 presidential election because it ignored a democratic process of religious governance.(I assume the author is connecting the "democratic process of religious governance" with megachurches, because the author notes that the Christianity of megachurches can make an all-out engagement in secular civil society.) Yet, this is wordy and not precise. Furthermore, the abstract does not highlight this thesis well.

All in all, there are enough grammatical issues in this paper that make it hard to know exactly what the thesis is. The grammatical issues contribute to an overall feeling of clumsy writing. 

Here is an example of a problematic sentence: "Even the narrow victory of the conservative regime implies that a socio-political influence of Protestants, especially megachurch con- gregants like SC, still remain active in Korea and further their Christianity can be politicized to win a ‘spiritual war’ against secular challenges at any time."

Is the author wanting to say: "Even the narrow victory of the conservative regime implies that the socio-political influence of Protestants, especially megachurch congregants like those who attend SC, still remains active in Korea, and furthermore their Christianity can be politicized to win a 'spiritual war' against secular challenges at any time."

There are many such sentences like this in this article. After figuring out the grammar, I am nonetheless left  wanting a more precise and organized delineation of the particular secular challenges that megachurch congregants are overcoming spiritually. Showing these would be a stronger and clearer way of showing how megachurches are solid socio-political actors  and not at all diminishing in their influence in their urban environments.

Author Response

I really appreciate your suggestions for making clear my thesis and correcting English words and sentences. I would be grateful if you give more suggestions for my revised manuscript, even though I tried to do my best to reflect almost all your suggestions literally. What I meant by  'secular challeges' can be considerd as a few examples such as quarantine measures related to the pandemic, the anti-discriminative law, and newly emerging sects as explained in this article. As you commented properly,  the meaning of religious governance can be defined broadly as a democratic process, but here it was used to mean a more practical and cooperative relationship between religion and government. In my opinion, there was no religous governance especially during the pandemic period in Korea.  Thank you for making a review in detail.